# Safety and Tolerability of COVID-19 Vaccine in Mast Cell Disorders Real-Life Data from a Single Centre in Italy

**DOI:** 10.3390/vaccines12020202

**Published:** 2024-02-16

**Authors:** Stefania Nicola, Marina Mazzola, Luca Lo Sardo, Erika Montabone, Iuliana Badiu, Federica Corradi, Maria Carmen Rita Azzolina, Maurizio Gaspare Dall’Acqua, Giovanni Rolla, Irene Ridolfi, Anna Quinternetto, Luisa Brussino

**Affiliations:** 1SCDU Immunologia e Allergologia, A.O. Ordine Mauriziano di Torino, C.so Re Umberto 109, 10128 Torino, Italy; stefania.nicola@unito.it (S.N.); marina.mazzola@unito.it (M.M.); llosardo@mauriziano.it (L.L.S.); erika.montabone@unito.it (E.M.); ibadiu@mauriziano.it (I.B.); fcorradi@mauriziano.it (F.C.); anna.quinternetto@unito.it (A.Q.); luisa.brussino@unito.it (L.B.); 2Department of Medical Sciences, University of Torino, C.so AM Dogliotti, 14, 10126 Torino, Italy; giovanni.rolla@unito.it; 3Health Direction, A.O. Ordine Mauriziano di Torino, C.so Re Umberto 109, 10128 Torino, Italy; mazzolina@mauriziano.it (M.C.R.A.);

**Keywords:** COVID-19, SARS-CoV-2, vaccine, safety, tolerability, premedication, hypersensitivity reaction, mastocytosis, mast cell disorders, allergy, epidemiology, risk assessment

## Abstract

**Background** In the past three years, COVID-19 has had a significant impact on the healthcare systems and people’s safety worldwide. Mass vaccinations dramatically improved the health and economic damage caused by SARS-CoV-2. However, the safety of COVID-19 vaccines in patients at high risk of allergic reactions still has many unmet needs that should be clarified. **Material and methods** A retrospective, single-centre study was performed by collecting demographic and clinical data of patients with Mast Cell Disorders (MCDs) to evaluate the safety and tolerability of COVID-19 vaccinations. Moreover, any changes in the natural history of the underlying disease following the vaccine have been evaluated. **Results** This study included 66 patients affected with MCDs. Out of them, 52 (78.8%) received a COVID-19 vaccination and 41 (78.8%) completed the vaccination course. Premedication came first in 86.6% of our patients. A total of seven (4.5%) patients complained about an immediate reaction and two (1.3%) had a late reaction. Worsening of MCD history was observed in a single patient. **Conclusions** Despite the overall high risk of allergic reactions, our study did not reveal any increased risk for SARS-CoV-2 allergic reactions in MCD patients, thus supporting the recommendation in favour of the SARS-CoV-2 vaccination. However, due to the potentially increased rate of anaphylactic reactions, MCD patients should receive vaccine premedication and should be treated in a hospital setting after an allergological specialistic evaluation.

## 1. Introduction

At the end of 2019, the coronavirus disease 2019 (COVID-19) emerged in China and rapidly spread into a global pandemic. It is caused by the severe acute respiratory syndrome coronavirus 2 (SARS-CoV-2) and is characterised by multisystemic involvement and high mortality [1].

Due to the lack of pre-existing natural immunity in human [2] or specific anti-COVID-19 drugs [3], at the very beginning, herd immunity had been proposed as a coping strategy. In spite of this, vaccines promptly became to be the most effective method to limit the pandemic spreading and protect the vulnerable. In this regard, a global immunisation campaign against SARS-CoV2 had been developed [4].

Since the beginning of the vaccination campaign, many new vaccines have been progressively authorised for use in the European Union: mRNA vaccines BNT162b2 (Pfizer-BioNTech^®^, Mainz, Germany) in 2020, mRNA-1273 (Moderna^®^, Madrid, Spain), viral vector vaccines ChAdOx1 nCoV-19 (Astra-Zeneca^®^, Nijmegen, Holland) and Ad26.COV2.S (Johnson & Johnson^®^, Beerse, Belgium) in 2021. Other vaccines have been licensed but not used for the vaccination of frail patients: VidPrevtyn Beta (Sanofi Pasteur^®^, Lyon, France), Nuvaxovid (Novavax^®^, Jevany, Czechia) and Bimervax (Hipra^®^, Amer, Spain) [5].

Vaccine-associated anaphylaxis is, in fact, rarely described in the general population, with an estimated incidence of one reaction per one million injections for most vaccines [6].

However, in the first few months, some cases of vaccines-related hypersensitivity reactions were observed [7] and worries about the safety of COVID-19 vaccines emerged [2]. This led to a slowdown of the vaccination process due to the suspected increased risk of anaphylactic reactions, especially in high-risk patients [8].

Hypersensitivity reactions are defined as an exaggerated or inappropriate response of the immunity against an antigen, and conventionally classified by Coombs and Gell [9] into four forms, according to the underlying molecular mechanism. Most reactions are mild and self-limiting; however, they could rarely be life-threatening or result in major complications, especially in high-risk patients (e.g., patients with mastocytosis) [10].

Concerning COVID-19 vaccines, recent evidence [11] also emphasizes the role of the contact system activation, through the “CARPA” pathway—Complement Activation-Related Pseudo Allergy—and the direct mast cell activation by the mRNA vaccine lipid nanoparticles.

Whatever the activation pathway, all organ systems could be involved, and symptoms are undistinguishable from those of an allergy. In spite of this, pseudoallergy can appear at first antigen exposure and the intensity in clinical manifestations usually decreases upon repeated exposures [12].

Due to the central role of mast cells in all the aforementioned hypersensitivity reaction pathways, it rapidly became clear that Mast Cell Disorders (MCDs) could have been part of the predisposing conditions for vaccine-related hypersensitivity reactions, despite the underlying mechanisms still remaining unknown [13].

Patients suffering from MCDs have an increased risk for anaphylactic reactions compared to the general population (up to 49%), mainly due to spontaneous degranulation of their clonal or abundant mast cells [14,15]. Anaphylaxis can be idiopathic or triggered by various elicitors. The most reported trigger of anaphylaxis in these patients is a Hymenoptera-venom sting; among others, many drugs, radio contrast media, alcoholic beverages, foods, and latex [16,17] are the most reported. In addition, some vaccines have also been reported as an enhancer of MCs activation or exacerbation of MCs mediator-derived symptoms [16,18].

As such, it is crucial to update the knowledge about the safety and tolerability of anti-SARS-CoV-2 vaccines in patients with a high risk of allergic reactions or anaphylaxis, like the ones suffering from Mast Cell (MC) Disorders (MCDs) [2,4,6].

There are currently few studies assessing the safety of a COVID-19 vaccination in patients suffering from MCDs. There are also partial and incomplete data on the need for any premedication of patients with MCDs undergoing COVID-19 vaccination, and the schedules to be used [17].

This study aimed to assess the prevalence of early and late COVID-19 vaccine-related reactions in patients affected by MCDs and evaluate any benefits from a premedication scheme. Moreover, a strict follow up of 12 months following the vaccination has been conducted to assess any possible worsening in the natural course of the underlying disease.

## 2. Materials and Methods

### 2.1. Study Design and Patients

This is a single-centrea retrospective single-arm study including all adult patients with MCDs attending the Immunology and Allergy University Unit at the Ordine Mauriziano Hospital, in Turin, Italy, from 1 December 2020 to 31 October 2023, who a received COVID-19 vaccination.

All of the MCDs patients that followed up at the immunology clinic were evaluated in the MCDs-dedicated outpatient’s clinic between three and one month before the COVID-19 vaccination.

According to SIAAIC/AAITO recommendation [19], all the patients underwent blood tests and instrumental exams as a part of routine follow up. In addition, at every follow-up visit, the mastocytosis activity score (MAS) [20] was stored along with information concerning the vaccination course and the premedication [21] recommended scheme (Table 1) was given.

The enrolled patients independently received the SARS-CoV2 vaccines, and data concerning the vaccinations were collected based on the clinical information at the follow-up on-site and remote visits and the patient’s medical history.

In the seven days after the vaccination, and after three months, all of the patients were called on the phone to assess either an immediate or late adverse reactions, or any changes in the natural history of the underlying disease. In case of any suspicion of a worsening in MCAD, the patient was referred for immediate clinical examination.

All of the enrolled patients released their written informed consent. The study was conducted by the Declaration of Helsinki and approved by the Institutional Review Board of Comitato Etico Territoriale Interaziendale (CET) “A.O.U. Città della Salute e della Scienza di Torino”, protocol code 0,073,320—dated 1 July 2022.

### 2.2. Mast Cells Disorders Evaluation

According to the most recent evidence [8], patients affected by MCD have been distinguished based on the diagnosis into neoplastic forms with clonal expansion of mast cells (mastocytosis belongs to this group), mast cell activation disorders (MCAD) which includes mast cell activation hyperplasia (MCAS) [22] and reactive mast cell hyperplasia.

Patients with mastocytosis were then subclassified into cutaneous mastocytosis—MC (maculo-papular cutaneous mastocytosis, diffuse cutaneous mastocytosis, and cutaneous mastocytoma), systemic mastocytosis—MS (bone marrow mastocytosis, indolent systemic mastocytosis, smoldering systemic mastocytosis, aggressive systemic mastocytosis, systemic mastocytosis associated with hematologic neoplasm and mastocytic leukemia), and mastocytic sarcoma [23].

### 2.3. Clinical Data Collection

For each patient, demographic characteristics, including sex and age, were collected. Moreover, data about age at diagnosis, any underlying disease according to WHO guidelines [24], history of anaphylaxis and its possible trigger, presenting signs—including cutaneous involvement, mast cells activation symptoms, anaphylaxis, haematological abnormalities, osteoporosis—, comorbidities, atopic status—including history of drug, food, Hymenoptera, latex, and inhalants allergy—, atopic dermatitis, allergic contact dermatitis, and ongoing treatment (H1-antihistamines (AH-1), H2-antihistamines (AH-2), montelukast, steroids, sodium cromoglycate, tyrosine kinase inhibitors, omalizumab) were recorded.

### 2.4. Laboratory Data Collection

Baseline serum tryptase level (Immunocap Fluorescence Enzyme Immunoassay Feia, Thermo Fisher Scientific, Uppsala, Sweden) was collected at least 48 h before the vaccination, as part of the routine workup. In case of any suspected immediate or delayed hypersensitivity reactions, the acute tryptase levels were also assessed no later than 120’ from symptoms onset. According to the manufacturer, tryptase levels below 10 mg/dL were considered normal.

In addition, the C-KIT D816V mutation status in the peripheral blood or bone marrow (BM), and findings at the BM biopsies (MC infiltrates, MC expressing CD2 or CD25) were also recorded for all the enrolled patients.

### 2.5. Mastocytosis Activity Score (MAS)

According to Siebenhaar F. et al. [20], the MAS was assessed in all the patients as a reported outcome. The severity of MCDs was considered as a mild disease activity in case of a MAS below 11.0, a moderate disease activity if MAS between 11.1 and 28.1, and a severe disease activity in case of a MAS upwards of 42.4 points.

All of the patients fulfilled the MAS questionnaire at every follow-up visit and every 7 days for 3 months after the vaccination.

### 2.6. Data Concerning COVID-19 Infections and Vaccines Received

Information about past SARS-CoV-2 infections and vaccinations were gathered through the analysis of data contained in the SIRVA Portal (Sistema Informativo Regionale per la gestione delle Vaccinazioni, released by Regione Piemonte) of the Piedmont region, and in digital medical records using the hospital application software, Babele WPF (v.1.0.0.2326), property of A.O. Mauriziano Umberto I—Turin. In detail, the data collected were vaccine type, vaccination setting, premedication scheme, possibly immediate or late reactions [25] and subsequent treatments.

The vaccines dosages used in this cohort of patients are the same as those used in the general population, in detail: 30 µg/0.3 mL for each dose for BNT162b2; 100 µg/0.5 mL for the first dose of mRNA-1273, reduced to a dosage of 50 µg/0.5 mL for the booster vaccination; a dosage of 2.5 × 10^8^ Inf.U (infectious units)/0.5 mL was used for ChAdOx1-S; and 8.92 log_10_ Inf.U/0.5 mL was administered for Ad26.COV2.S vaccines.

Moreover, information concerning the vaccination course was recorded. A vaccination course is considered complete if it comprehends three doses or more of a microRNA vaccine. Alternatively, one dose of Ad26.COV2.S and a microRNA-vaccine booster dose, two doses of ChAdOx1 and a microRNA-vaccine booster, or two doses of mRNA-vaccine and COVID-19 are equally considered.

### 2.7. Immediate or Delayed Reactions

The most common reactions in response to vaccinations are type I and type IV hypersensitivity [9].

Type 1 reaction is IgE mediated, and its clinical manifestations are consequent to mast cells and basophils activation. It occurs after several minutes or up to 6 h after allergen exposure; therefore, it is usually called “immediate reaction”.

Type IV hypersensitivity occurs within 2–3 days of a subsequent exposure to the antigen, as result of CD4+ and CD8+T cells activation by dendritic cells presenting antigens. This is the representative mechanism of the “delayed or late-onset reactions”, whose symptoms may include urticaria, eczema, erythema, or itching for hours to days after the vaccine injection [26].

Also, local injection site reactions have been described several days after immunisation, and many authors classified them as type IV reactions [27].

Moreover, non-allergic adverse reactions following vaccine administration were collected on a weekly basis for one month; local reactions, intended as pain, erythema, and swelling at injection site, fever, headache, myalgias, arthralgias, lymphadenomegaly, flu-like symptoms, abdominal pain and fatigue were recorded.

### 2.8. Worsening of MCDs Evaluation after COVID-19 Vaccines

The suspicion of a MCDs worsening after a vaccination course was raised based on the increase in the patient-reported outcome. In this case, the patient was immediately referred to the immunology clinic for an urgent evaluation, and a complete re-evaluation workup had been made.

### 2.9. Statistical Analysis

The anonymised data were entered in IBM SPSS Statistics for Windows, version 26 (IBM Corp, Armonk, NY, USA) and a statistical analysis was executed. The Kolmogorov–Smirnov normality test was used to calculate the normal distribution of data; subsequently, a descriptive analysis of the variables was performed.

The assessment of baseline characteristics in the whole cohort was established, then described as mean for continuous variables and as absolute and relative frequencies for categorical variables.

## 3. Results

### 3.1. Features of the Enrolled Cohort

We enrolled 66 patients (37 females, 56%) affected with MCDs, with a median age at diagnosis of 52.9 years (range 2–90). A total of twenty-seven (40.9%) patients were diagnosed with Indolent Systemic Mastocytosis (ISM), three (4.5%) had Systemic Mastocytosis with an associated haematological neoplasm (SM-AHN), twelve (18.2%) had a diagnosis of Cutaneous Mastocytosis (CM), and twenty-four (36.4%) suffered from Mast Cell Activation Syndrome (MCAS). The distribution of MCDs diagnosis among the enrolled cohort is shown in Figure 1.

At the clinical evaluation before the vaccination, the mean basal serum tryptase (BST) was 54.5 ng/mL (range 3.9–200 ng/mL) and the mean MAS was 10.89 (±2.78). The distribution of those levels among each Mast Cell Disease subtype is represented in Figure 2.

Regarding the atopic status, 34 (51.5%) patients had a history of allergies. Specifically, eleven (16.6%) of them suffered from a respiratory allergy, primarily due to perennial antigens (mainly house dust mites); out of these patients, eight had allergic rhinitis, and three had atopic asthma. In addition, ten (15.2%) patients had a food allergy history (one red meat, three hazelnuts, one peanut, two shrimp, one fish, and two peach allergies), while 17 (25.7%) suffered from Hymenoptera venom allergy (eleven *vespidae*, six *apidae*) and one with a latex allergy. Lastly, five (7.6%) had a history of drug allergy; out of them, one patient had a proven allergy to polyethylene glycol (PEG). Allergic contact dermatitis affected 4.5% of our cohort and atopic dermatitis was present in one patient. Moreover, thirty-one (46.9%) patients had a history of anaphylaxis, mostly (64.5%) due to Hymenoptera venom sting. As well, three (9.7%) patients experienced idiopathic anaphylaxis. No patients showed an anaphylactic reaction after previous vaccinations (Table 2).

### 3.2. SARS-CoV-2 Vaccination Course in the MCDs Cohort

Among the enrolled patients, 14 patients refused the vaccination, whereas 52/66 (78.8%) received at least one dose of a COVID-19 vaccination. Out of the fifty-two vaccinated patients, forty-one (78.8%) completed the vaccination course, nine (17.3%) received two doses, and two patients performed a single COVID-19 vaccine administration.

Out of 52 vaccinated patients, 49 (94.2%) were fully vaccinated with mRNA-vaccine, 39 (75%) of them with BNT162b2 exclusively, and the remaining 10 were vaccinated with at least one dose of mRNA-1273. A total of two patients had the first ChAdOx1, then two doses of mRNA-1273. To note, the PEG-allergic patient received the vaccination with Ad26.COV2.S. During the study period, 157 doses of COVID-19 vaccine were overall administered.

All of the patients were vaccinated in a hospital setting.

The vaccine has been administered intramuscularly on the deltoid muscle by trained nurses, who were supported by a skilled resuscitation staff. Every patient was kept on a 60 min observation with systematic monitoring of vital signs after the vaccination.

### 3.3. Medications Taken before the Vaccination

Concerning the premedication scheme, 18 (34.6%) patients pursued the usual daily antihistamine-1 (AH-1) treatment; 18 (34.6%) patients started AH-1 a few days before the administration and continued it for five days afterwards; five (9.6%) doubled the AH-1 daily dose. Only seven (13.5%) patients underwent vaccination without any premedication. A total of four (7.7%) patients underwent vaccination following other premedication schemes: one patient continued AH-1 home treatment and added Montelukast 2 h before vaccination; another one doubled AH-1 treatment and added famotidine and montelukast, lastly one patient doubled the AH-1 treatment and added sodium cromoglycate. A total of seven (13.5%) patients underwent a COVID-19 vaccination without any AH-1 premedication.

### 3.4. Immediate SARS-CoV-2 Vaccine Reactions

Of the over 157 administered vaccine doses, seven (4.5%) immediate reactions have been reported, which occurred in four (7.7%) patients.

In detail, a woman affected with monomorphic maculopapular cutaneous mastocytosis, without any previous history of anaphylaxis, showed facial flushing, cough and vomiting a few minutes after the second BNT162b2 vaccine administration, despite AH-1 premedication. The patient was treated with 125 mg intravenous (IV) methylprednisolone, 10 mg IV chlorpheniramine and inhaled SABA, with prompt clinical improvement. This reaction was compatible with a level 2 anaphylaxis, according to the Brighton criteria, even though epinephrine was not administered. After that, the patient was referred to our allergy clinic, and skin tests for the BNT162b2 vaccine and PEG were performed, with negative results. No absolute contraindications for future vaccinations were made, but the patient refused it.

A patient with asthma (step 3 GINA, regularly treated with beclomethasone/formoterol 100 mcg/6 mcg, twice daily) experienced an immediate exacerbation of respiratory symptoms after the two vaccine doses he received, both resolved with a short-acting beta-agonist (SABA) dose. Moreover, a premedicated CM patient experienced an immediate flushing shortly after the first BNT162b2 dose. She was successfully treated with oral AH-1. The patient underwent the second vaccination with the BNT162b2 vaccine without experiencing any adverse effects. Lastly, a woman affected with mMCAS with a previous anaphylaxis history developed a fugax and widespread urticaria following every vaccine administration and despite AH-1 premedication.

### 3.5. Delayed SARS-CoV-2 Vaccines Reactions

Concerning the delayed reactions that occurred in our cohort, we reported a sine materia and localised itch after the third dose of the BNT162b2 vaccine, which was responsive to AH-1, and a case of delayed disseminated wheals after the first dose in a non-premedicated patient (Table 3).

### 3.6. Safety of COVID-19 Vaccination in MCDs Patients

No adverse reactions were observed in patients not taking any AH-1 premedication. No immediate or delayed vaccination-related deaths or hospitalisation requiring severe adverse reactions were detected.

### 3.7. Non-Allergic Adverse Reactions to SARS-CoV-2 Vaccines

Any adverse reaction following each vaccine administration were recorded every seven days for the first month.

In our cohort, flu-like symptoms, such as myalgias, arthralgias, and fatigue were recorded in seven vaccinated patients and fever occurred in fourteen patients; both of these instances were treated with paracetamol or non-steroidal anti-inflammatory drugs (NSAIDs), with a complete clinical remission in about 72 h. The six site injection reactions, characterised by local pain and swelling, were described and resolved in 5 days without any therapeutic measures. Overall, two patients complained of headache that arose within hours after vaccination and lasted for 48 h. A single patient reported diarrhea and abdominal pain. A case of unilateral axillary lymphadenopathy near the injection site was described with a full resolution in about 2 weeks without any medications.

### 3.8. Worsening of the Natural History of the Disease

Concerning the assessment of possible changes in the natural course of MCAD, only one worsening of the clinical conditions was observed. After three months from the third BNT162b2 vaccine dose, a patient affected with ISM started experiencing recurrent MC-activation symptoms, including facial flushing, widespread itching, and headache, and his MAS was 36.7. Therefore, a detailed workup panel was performed: high serum tryptase level (194 μg/L) and diffuse skeletal involvement with large osteolytic lesions were detected. In addition, the bone marrow biopsy reflected a high MC burden. For these reasons, the diagnosis of aggressive systemic mastocytosis (ASM) was conducted. The patient was referred to a hematologist and midostaurine was started, and a significant clinical improvement was observed within a few months.

## 4. Discussion

This is the first study assessing both immediate and delayed reactions, as well as worsening in the underlying disease in a quite high number of patients with MCDs [28].

In our MCAD cohort, the rate of immediate hypersensitivity reactions after the COVID-19 vaccination was low (4.5%), despite it being even greater than shown in a recent study of highly allergic non-mastocytosis patients [29]. This was probably related to the characteristics of our population, that included patients with a high risk of anaphylaxis, mainly because of their underlying MCs disease, in addition to their highly atopic status.

On the other hand, our results are comparable with the largest study published to date on this topic in patients with MC activation disorders, which shows a prevalence of adverse reactions of 6% [30].

In our cohort, the observed reactions following a COVID-19 vaccination were usually mild and involved only one organ, mainly the skin or respiratory tract.

Moreover, the rate of more severe adverse reactions involving two or more systems in our cohort is 1.9%, comparable to the 1.8% described by Giannetti MP et al., in patients with Mast Cell Activation Disorders, although higher than the one of the general populations [30].

The prevalence of delayed reactions due to a COVID-19 vaccination in our cohort was 3.8%; this percentage was significantly lower than the one published in the Shavit et al. study, which reported 14.7% [29]. This difference may be due to the allergy pre-vaccination screening, in which long-term antihistamine premedication was recommended. Nevertheless, we cannot exclude this data, which could be influenced by the low-numbered sample size of the study.

Since the only well-known exception to vaccination is the allergy against a constituent of the vaccine [4], our PEG-allergic patient underwent COVID-19 vaccination with Ad26.COV2.S, containing Polysorbate 80 as an excipient and no drug-related adverse events were observed.

As pointed out by Rama et al. no controlled studies concerning premedication protocols in patients affected by Mast Cell Disorders have been conducted, so that the premedication scheme is not standardised, and its effectiveness is unknown. In spite of that, vaccination should be considered as well as other invasive procedures (minor and major surgery, general anesthesia, and radiological testing with radiocontrast media) and experts suggest pre-medications with H1 and H2-antihistamines, leukotriene blockers and glucocorticoids. The European Competence Network on Mastocytosis and the American Initiative in Mastocytosis recommend the use of premedication with an H1 antihistamine 30–60 min before a COVID-19 vaccination [31].

In our cohort, the most employed premedication scheme was based on AH-1 (59.6%). Almost one-third (32.6%) of the patients underwent vaccination without any premedication and no differences were found in terms of reactions compared with patients who did not take any medication before. According to the European Competence Network on Mastocytosis and the American Initiative in Mast Cell Disease, patients with Mastocytosis at high risk for anaphylaxis should premedicate with H1-antihistamine 30 to 60 min before the vaccination. Until randomised prospective trials on the need for AH premedication are performed, it is suggested that patients pursue the daily AH treatment or start it a few days before the vaccination [14].

Thus, our analysis suggests that both mRNA-based and adenoviral vector COVID-19 vaccines appear to be safe and well tolerated in a population of highly allergic MCD patients. However, the small sample size of our cohort limited this study and did not allow us to identify rare events and thus affected the generalisability of the data.

The main strength and, at the same time, the main limitation of this single-centre study is its real-life nature. Major limitations include the retrospective design, the heterogeneity of the cohort, and the differences in the premedicating scheme. In addition, all of the vaccinations occurred in a hospital setting, where many experts in the management of immediate allergic reactions are usually present.

Regarding the possible worsening of the natural history of MCAD, we reported exclusively one case of ASM diagnosed after the vaccination in a patient previously affected with ISM. Due to the paucity of numbers, no statistical conclusions can be drawn.

Because of the rapid spread of COVID-19, more and more evidence to ensure the safety and tolerability of COVID-19 vaccines in MCD patients is needed; although, currently the risk of anaphylaxis in this population seems to be low. Therefore, patients with MDCs should not be excluded from the mass vaccination campaign [32].

Nevertheless, due to the overall high risk of allergic reactions, an experienced allergist team represents an effective strategy to ensure MCDs patients the best clinical practice, identifying the best vaccines setting and the subsequent management [21,33].

Thus, all patients with high allergy risk should be encouraged to undergo vaccination, starting the suggested premedication in addition to the daily medications, and should be educated about the need to carry along the epinephrine auto-injector, which use should be simulated at each allergological specialist visit or epinephrine represcription [34].

## 5. Conclusions

In conclusion, our findings suggest that patients suffering from MCDs can be safely vaccinated against SARS-CoV-2, even those with anaphylactic history and/or proven allergy to the vaccine components. In this regard, MCDs do not represent a contra-indication to a COVID-19 vaccination, but the mentioned precautions should be observed.

Moreover, our study underlined the importance of a qualified team for clinical evaluation of the allergic risk to COVID-19 vaccines, the eligibility for vaccination, and the need for any premedication, that minimizes the risk for vaccine induced allergic reaction.

## Figures and Tables

**Figure 1 vaccines-12-00202-f001:**
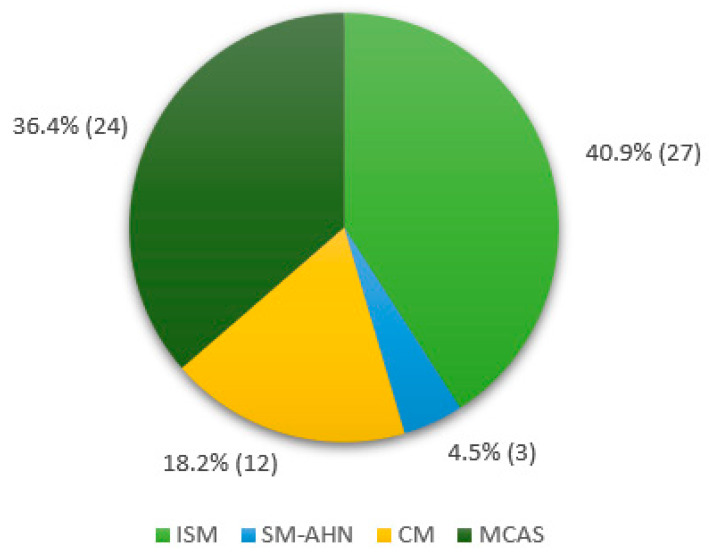
Distribution of MCD diagnosis in the enrolled cohort % (n). Abbreviations: ISM, Indolent Systemic Mastocytosis; SM-AHN Systemic Mastocytosis with an associated haematological neoplasm; CM, Cutaneous Mastocytosis; MCAS, Mast Cell Activation Syndrome.

**Figure 2 vaccines-12-00202-f002:**
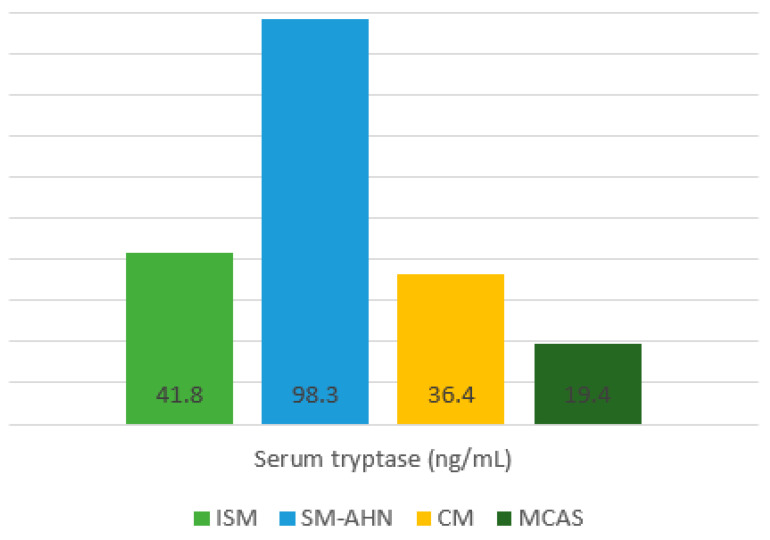
Mean basal serum tryptase level (ng/mL). Abbreviations: ISM, Indolent Systemic Mastocytosis; SM-AHN Systemic Mastocytosis with an associated haematological neoplasm; CM, Cutaneous Mastocytosis; MCAS, Mast Cell Activation Syndrome.

**Table 1 vaccines-12-00202-t001:** Abbreviations: MC, mast cell; PEG, polyethylene glycol.

	Premedication Scheme	Vaccination Setting	Supervision after Vaccination	Carrying Adrenaline Autoinjector at Vaccination Site
High-risk - Previous hypersensitivity reactions to vaccination - Severe uncontrolled MC activation symptoms	H1-antihistamine (i.e. Cetirizine) one day to 5 days after vaccination	Hospital setting in presence of resuscitation emergency team	Up to 4 h	Yes
Mild-risk - Anaphylactic reactions with many triggers - Idiopathic anaphylaxis - Well-controlled MC activation symptoms - Mild reaction to previous vaccinations	H1-antihistamine (i.e. Cetirizine) one day to 5 days after vaccination	Hospital setting	Up to 60 min	Yes
Low-risk - No previous anaphylaxis - Anaphylaxis to one and well-defined trigger - No previous reactions to PEG	H1-antihistamine (i.e. Cetirizine) one day to 5 days after vaccination	Outpatient setting	Up to 60 min	Yes

**Table 2 vaccines-12-00202-t002:** Characteristics of the enrolled cohort.

Patients (n = 66) and Features	
Female, n (%)	37 (56%)
Age, mean (range)	52.9 year (2–90)
Serum tryptase level, mean (range)	54.5 ng/mL (3.9–200)
Atopy, n (%)	34 (51.5%)
Respiratory allergy, n (%)	11 (16.6%)
Food allergy, n (%)	10 (15.2%)
Hymenoptera venom allergy, n (%)	17 (25.7%)
Latex allergy, n (%)	1 (0.66%)
Drug allergy, n (%)	5 (7.6%)
Polyethylene glycol (PEG) allergy, n (%)	1 (0.66%)
History of anaphylaxis, n (%)	31 (46.9%)

**Table 3 vaccines-12-00202-t003:** Description of immediate and delayed adverse reactions to COVID-19 vaccination.

**IMMEDIATE REACTIONS**						
**PATIENT**	**DIAGNOSIS**	**VACCINE**	**DOSE**	**PREMEDICATION**	**SYMPTOMS**	**THERAPY**
**1**	MCAS	BNT162b2	II	Yes	Anaphylaxis	CS, AH1, SABA
**2**	ISM	BNT162b2	I, II	Yes	Asthma exacerbation	SABA
**3**	CM	BNT162b2	I	Yes	Flushing	AH1
**4**	MCAS	BNT162b2	I, II, III	Yes	Acute urticaria	-
**DELAYED REACTIONS**						
**PATIENT**	**DIAGNOSIS**	**VACCINE**	**DOSE**	**PREMEDICATION**	**SYMPTOMS**	**THERAPY**
**1**	CM	BNT162b2	I	No	Acute urticaria	-
**2**	CM	BNT162b2	III	Yes	Itch	AH1

Abbreviations: CS, corticosteroids; AH1, H1-antihistamines; SABA, short-acting beta agonists.

## Data Availability

The data that support the findings of this study are available from the last author, [L.B.], upon reasonable request.

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
