# Peer review of "Safety and Tolerability of COVID-19 Vaccine in Mast Cell Disorders Real-Life Data from a Single Centre in Italy"

_vaccines, 2024, doi:10.3390/vaccines12020202_

Round 1

Reviewer 1 Report

Comments and Suggestions for Authors

1)      In my opinion, as a non-expert in the field of allergies and related events, a brief description of MCD compared to a “regular allergy” should be include, specially since the authors state  In the discussion that “In our MCAD cohort, the rate of immediate hypersensitivity reactions after the  COVID-19 vaccination was low (4.5 %), despite it being even greater than shown in a recent study on highly allergic non-mastocytosis patients29 .”

2)      The number of patients does not appear in the material and methods

3)      There is not a control group (determination of post vaccine effects in patients without MCD)

4)      The distribution of the severity of MCD is not shown in a table or throughout the manuscript.

5)      In my opinion from the line 201 to 207 should be  in the material and methods

6)      “In our cohort, flu-like symptoms, as myalgias, arthralgias, and fatigue were recorded  in 7 vaccinated patients and fever occurred in 14 patients; both of them were treated with  paracetamol or non-steroidal anti-inflammatory drugs (NSAIDs), with complete clinical  remission in about 72 hours. Six site injection reactions, characterized by local pain and swelling were described and resolved in 5 days without any therapeutic measures. Two  patients complained of headache, arised within hours after vaccination and lasted for 48  hours. A single patient reported diarrhea and abdominal pain. A case of unilateral axillary  lymphadenopathy near the injection site was described with a full resolution in among 2  weeks without any medications.”

Are these symptoms different from those of the general population? In order to the Author to answer this question a control group is required!!!

7)      Table 2 summarize the adverse effects in 4 patients (immediate reactions) and 2 patients (delayed reactions). Are Patients 1 and 2 the same individuals in early and late reactions? If the answer is Yes, please may you describe any particular or relevant information of these patients, by the way is the number of patients with adverse effect significant or not significant respect to total number of patients who participate in the study? and what kind of vaccine did these patients receive?

Author Response

We are extremely grateful for the comments that the reviewer made us, and the criticisms he highlighted.

Here you have the point-by point reply:

1) In my opinion, as a non-expert in the field of allergies and related events, a brief description of MCD compared to a “regular allergy” should be include, specially since the authors state  In the discussion that “In our MCAD cohort, the rate of immediate hypersensitivity reactions after the  COVID-19 vaccination was low (4.5 %), despite it being even greater than shown in a recent study on highly allergic non-mastocytosis patients29 .”

Thank you for your suggestions. In the revised manuscript, we added a brief sentence concerning the increased risk of allergic reactions in MCAD patients (Lines 71-72)

2) The number of patients does not appear in the material and methods

Thank you for your remark. We used to list the number of patients in the “Result” section. Hope this could not affect the overall understanding of the manuscript.

3) There is not a control group (determination of post vaccine effects in patients without MCD)

Thank you for your comment. This is a retrospective cohort study, so that we could not have a proper control group. Thus, we referred to general population data already described in literature about the safety and the tolerability of COVID-19 vaccines..

4) The distribution of the severity of MCD is not shown in a table or throughout the manuscript.

Thank you for your advice. The distribution of the MCD patients, according to their diagnosis and severity is represented in Figure 1 and described in the manuscript (line 203-207).

5) In my opinion from the line 201 to 207 should be  in the material and methods

Thank you for your advice. We used to place these data in the “Result” section. Hope this could not affect the overall understanding of the manuscript.

6) “In our cohort, flu-like symptoms, as myalgias, arthralgias, and fatigue were recorded  in 7 vaccinated patients and fever occurred in 14 patients; both of them were treated with  paracetamol or non-steroidal anti-inflammatory drugs (NSAIDs), with complete clinical  remission in about 72 hours. Six site injection reactions, characterized by local pain and swelling were described and resolved in 5 days without any therapeutic measures. Two  patients complained of headache, arised within hours after vaccination and lasted for 48  hours. A single patient reported diarrhea and abdominal pain. A case of unilateral axillary  lymphadenopathy near the injection site was described with a full resolution in among 2  weeks without any medications.”

Are these symptoms different from those of the general population? In order to the Author to answer this question a control group is required!!!

6) Thank you for the comment. Our cohort shows a prevalence of vaccination adverse effect similar to the general population. Please find this information on EMA report (https://www.ema.europa.eu/en/human-regulatory-overview/public-health-threats/coronavirus-disease-covid-19/covid-19-medicines/safety-covid-19-vaccines) and paper about vaccine surveillance in general population (Tsang RS, Joy M, Byford R, et al. Adverse events following first and second dose COVID-19 vaccination in England, October 2020 to September 2021: a national vaccine surveillance platform self-controlled case series study. Euro Surveill. 2023;28(3):2200195. doi:10.2807/1560-7917.ES.2023.28.3.2200195)

7) Table 2 summarize the adverse effects in 4 patients (immediate reactions) and 2 patients (delayed reactions). Are Patients 1 and 2 the same individuals in early and late reactions? If the answer is Yes, please may you describe any particular or relevant information of these patients, by the way is the number of patients with adverse effect significant or not significant respect to total number of patients who participate in the study? and what kind of vaccine did these patients receive?

Thank you for these extremely interesting pointing note. The patients are actually different. The number of adverse effects is low, and similar to other comparable cohorts. That means, COVID-19 vaccines are well-tolerated in MCAS patients.

The type of administered vaccines is shown in Table 2 and expressed in the manuscript.

Reviewer 2 Report

Comments and Suggestions for Authors

The manuscript entitled Safety and Tolerability of COVID-19 Vaccine in Mast Cell 2 Disorders Real-Life Data from a Single Centre in Italy presents an interesting piece of work in the evaluation of Anaphylactic reaction leading to MCD in COVID-19 vaccine receipient from December 2020 to 31st October 2023. I have few concerns that needs to be addressed for making this article best choice for the readers"

1. The authors can represents the results of MCDs in tabular or pictorial to make it more attractive as no Fig. is present in the manuscript.

2. Did the authors tested for COVID-19 antibodies in MCDs patients. Presuming it will be lower than healthy individuals, but we don't have the right answer for you r settings.

3. The authors observe the anaphylactic response only with in 60 minutes. what are the long time response from those individuals?

4.Regarding the possible worsening of the natural history of MCAD, we reported ex clusively one case of ASM diagnosed after the vaccination in a patient previously affected with ISM. Due to the paucity of numbers, no statistical conclusions can be drawn- This case is major drawback and we can't say s "Despite the overall high risk of allergic reactions, our 24 study suggests that MCD patients can be safely vaccinated against SARS-CoV-2"

Please modify the sentance.

Author Response

We are extremely grateful for the comments that the reviewer made us, and the criticisms he highlighted.

Here you have the point-by point reply:

The manuscript entitled Safety and Tolerability of COVID-19 Vaccine in Mast Cell 2 Disorders Real-Life Data from a Single Centre in Italy presents an interesting piece of work in the evaluation of Anaphylactic reaction leading to MCD in COVID-19 vaccine receipient from December 2020 to 31st October 2023. I have few concerns that needs to be addressed for making this article best choice for the readers"

Thank you!

  1. The authors can represents the results of MCDs in tabular or pictorial to make it more attractive as no Fig. is present in the manuscript.

Thank you for the suggestion, you’ll find tables and figures as a n attached supplementary file.

  1. Did the authors tested for COVID-19 antibodies in MCDs patients. Presuming it will be lower than healthy individuals, but we don't have the right answer for you r settings.

Thank you for your suggestion. Therefore, we did test Ab anti-SARS-CoV-2 in our cohort, and the results did not differ from the general population. However, the results fell outside from the scope of the manuscript, and we decided not to include them in the paper.

  1. The authors observe the anaphylactic response only with in 60 minutes. what are the long time response from those individuals?

Most immediate IgE-mediated reactions, as anaphylaxis, mainly occur within one hour, even if rare allergic reactions occur up to six hours. Possible allergic reactions arose after this time frame have been assessed by phone interview.

  1. Regarding the possible worsening of the natural history of MCAD, we reported ex clusively one case of ASM diagnosed after the vaccination in a patient previously affected with ISM. Due to the paucity of numbers, no statistical conclusions can be drawn- This case is major drawback and we can't say s "Despite the overall high risk of allergic reactions, our 24 study suggests that MCD patients can be safely vaccinated against SARS-CoV-2". Please modify the sentance
  2. Thank you for the comment. The sentence has now been changed. (Lines 24-26)

For now, it is not possible conclude anything about the risk of worsening and progression of the disease following vaccination because of the paucity of the cohort and the short follow-up period.